# SaulLM-54B & SaulLM-141B: Scaling Up Domain Adaptation for the Legal Domain

**Pierre Colombo**
equal
Equall
MICS - CentraleSupelec

**Telmo Pires**
equal
Equall

**Malik Boudiaf**
equal
Equall

**Rui Melo**
equal
Equall

**Dominic Culver**
equal
Equall

**Etienne Malaboeuf**
CINES

**Gabriel Hautreux**
CINES

**Johanne Charpentier**
CINES

**Michael Desa**
equal
Equall

## Abstract

In this paper, we introduce `SaulLM-54B` and `SaulLM-141B`, two large language models (LLMs) tailored for the legal sector. These models, which feature architectures of 54 billion and 141 billion parameters, respectively, are based on the Mixtral architecture. The development of `SaulLM-54B` and `SaulLM-141B` is guided by large-scale domain adaptation, divided into three strategies: (1) the exploitation of continued pretraining involving a base corpus that includes over $540$ billion of legal tokens, (2) the implementation of a specialized legal instruction-following protocol, and (3) the alignment of model outputs with human preferences in legal interpretations. The integration of synthetically generated data in the second and third steps enhances the models' capabilities in interpreting and processing legal texts, effectively reaching state-of-the-art performance and outperforming previous open-source models on `LegalBench-Instruct`. This work explores the trade-offs involved in domain-specific adaptation at this scale, offering insights that may inform future studies on domain adaptation using strong decoder models. Building upon `SaulLM-7B`, this study refines the approach to produce an LLM better equipped for legal tasks. We are releasing base, instruct, and aligned versions on top of `SaulLM-54B` and `SaulLM-141B` under the MIT License to facilitate reuse and collaborative research.

## 1 Introduction

LLMs have demonstrated exceptional capabilities across various domains [1, 67, 60, 76, 39, 40, 77, 7, 28], excelling in tasks such as language translation [6], medical diagnostics [16, 11, 12], and automated code generation [4, 42, 32], among others. These achievements highlight the potential for human-like communication through large language models (LLMs). Despite the significant potential benefits, the adaptation of most recent LLMs for legal tasks has not been extensively examined, with only two recent studies cited from [18, 55, 86], and its impact on society could be substantial. Indeed, at a time when legal systems in many countries are overburdened [71], the development of robust and high-performing legal LLMs could provide critical support to lawyers and judicial systems [65, 10]. However, adapting LLMs to the legal domain presents unique challenges, particularly because of the vast scale involved, with hundreds of billions of existing legal data available.

38th Conference on Neural Information Processing Systems (NeurIPS 2024).

Previous efforts to tailor LLMs to the legal sector have encountered significant challenges [18, 55, 86]: first, a limited model scale, capped at 7/12B parameters, which is considerably smaller than the largest open-source models [7, 40]; second, training datasets restricted to no more than 30 billion tokens, significantly fewer than potentially available tokens [30, 54]. Given the importance of scale and breadth in effectively adapting LLMs to new domains, this paper aims to answer the following research question:

*How much can we improve the specialization of generic LLMs for legal tasks by scaling up both model and corpus size?*

In this paper, we present an empirical study on the scalability and domain adaptation of LLMs in the legal sector. Relying on a corpus exceeding 500B tokens and models up to 141B parameters, our research seeks to address the gaps in the examination of legal applications. A novel aspect of our approach is the adaptation of large-scale Mixture of Experts (MoE) models with 54B and 141B parameters, which have gained significant traction in recent months [93, 22, 45, 90, 87, 61]. Formally, this study makes two principal contributions:

**1. Comprehensive Analysis of Domain Adaptation Strategies for Legal LLMs** Domain adaptation for legal LLMs remains a challenging and somewhat underexplored area. This work advances the field by specializing each step in the process of developing modern LLMs, from continued pretraining to instruction fine-tuning and alignment, relying on both synthetic and real data. This paper offers a fresh perspective on the efficacy of each step and its value for adapting to the legal domain, potentially guiding further research in the legal domain as well as in other expert domains.

**2. `SaulLM-54B` & `SaulLM-141B`: Joining SaulLM-7b to form a Family of Legal LLMs under Permissive License**[1] We specialize general-purpose, large-scale LLMs for the law. This work represents an ambitious advancement in terms of scale and leveraging the increasingly popular MoE architecture. While this architecture is widely used, its specific applications within focused domains, particularly the legal sector, are still largely unexplored. By releasing these models, we aim to foster further research in legal NLP and contribute to unlocking the full potential of LLMs.

## 2 Related Work

### 2.1 Domain Specialization For Large Language Models

The process of domain specialization for LLMs has demonstrated promising results in areas such as medicine [16], science [74], translation [6, 5], or code [66, 42, 4]. Models like SciBERT [9], PubMedBERT [75], Galactica [74] and Meditron [16] have been specifically trained on domain-related corpora to enhance their performance. Studies have identified that both the scale of the model and the size of the in-domain data are crucial for achieving strong domain adaptation [16, 66].

In the legal domain, earlier models such as LegalBERT [15], InCaseLawBERT [59], and SaulLM-7B [18], among others, while pioneering, have been constrained by their relatively small scale and the specificity of their training data, which covers a limited number of documents and jurisdictions. Our work aims to build on these efforts by deploying LLMs at an unprecedented scale, utilizing models of up to 141B parameters and a base corpus exceeding 500 billion tokens to significantly enhance the depth and breadth of legal language comprehension and generation.

### 2.2 Legal Domain Adaptation for Modern LLM

The field of legal domain adaptation has traditionally concentrated on refining models through pretraining on specialized corpora [15, 18, 21]. Yet, in the current paradigm, pretraining represents just one aspect of the solution, as LLMs often utilize techniques like instruction fine-tuning and alignment, employing algorithms such as DPO [63], PPO [68] or RLHF [57, 44].

Recent domain-adapted models, such as SaulLM or Legal-FLAN-T5 (a closed model), have tried to improve alignment with legal instructions. However, SaulLM is a smaller model, and Legal-FLAN-T5, is based on an outdated architecture and does not leverage the extreme scale pretraining that contemporary models do. Moreover, it not being publicly available stymies progress vital for advancing research and applications in the legal sector.

---

[1]Model will be made available at `https://huggingface.co/`.

We believe this work pioneers a holistic approach to domain adaptation by training modern LLMs specifically for the legal domain, from pretraining to instruction fine-tuning and legal preference alignment. We demonstrate that synthetic data can be effectively utilized for alignment, advancing beyond SaulLM-7B's use solely of instruction fine-tuning. The resulting models, `SaulLM-54B` and `SaulLM-141B`, lay the groundwork for further research and development, and expand access to high-performance legal LLMs.

## 3 Data Collection and Corpus Construction

This section outlines our approach to assembling and refining a comprehensive legal text corpus tailored for training large language models in the legal domain.

### 3.1 Pretraining Corpora

The diversity of legal systems worldwide, from common law to civil law traditions, presents unique challenges and opportunities [54, 35]. To address this, we compiled an extensive English-language corpus from various jurisdictions including the U.S., Europe, Australia, and others [3, 33], which comprises 500 billion tokens before cleaning and deduplication.

#### 3.1.1 Legal Sources

Our base corpus combines various legal datasets [53] with newly sourced public domain documents. It includes significant collections such as the FreeLaw subset and the MultiLegal Pile, augmented with extensive web-scraped content. Table 1 summarizes the composition and scale of our dataset.

#### 3.1.2 Other Sources

**Replay Sources.** To mitigate the risk of catastrophic forgetting during model training, we reintroduced data from earlier training distributions [48, 16, 73, 72, 25, 24]. This replay strategy incorporates general data sources such as Wikipedia, StackExchange, and GitHub, and makes up approximately $2\%$ of the total training mix. These datasets are sampled from SlimPajama [69, 19, 70].

Additionally, we included $5\%$ of math datasets in the pretraining mix using commercially available math sources. We found this approach usefull for retaining the reasoning performance of the final model and avoiding the weaker performance observed in previous research attempts like SaulLM-7B[2].

Table 1: **Sources of Legal Pretraining Data**

| Source Name | Tokens (B) |
|---|---|
| FreeLaw Subset from The Pile | 15 |
| EDGAR Database | 5 |
| English MultiLegal Pile | 50 |
| English EuroParl | 6 |
| GovInfo Statutes, Opinions & Codes | 11 |
| Law Stack Exchange | 0.019 |
| Comm Open Australian Legal Corpus | 0.5 |
| EU Legislation | 0.315 |
| UK Legislation | 0.190 |
| Court Transcripts | 0.350 |
| UPSTO Database | 4.7 |
| Web Data (legal) | 400 |
| Other | 30 |
| **Total** | **520** |

**Instruction Sources.** We found that incorporating conversational data during pretraining is advantageous, drawing inspiration from recent breakthroughs in neural machine translation [6]. Studies suggest that the enhanced translation capabilities of LLMs can be attributed to the presence of accidental parallel data within their training corpora. Accordingly, we have integrated the Super Natural Instruction [83] and FLAN collection [46] into our pretraining mix, enriching the dataset with diverse instructional content.

**Data for Model Annealing.** Model annealing is primarily achieved through a methodical reduction of the learning rate [58, 38], known as learning rate annealing.

In our experiments, model annealing with high-quality, domain-relevant data significantly enhanced performance. Conversely, repetitive synthetic data from initial instruction fine-tuning harmed performance. Therefore, we used the commercial portion of the LawInstruct dataset for model annealing,

---

[2]These findings also align with the high percentage of math and STEM in the pretraining mix from [56].

which proved more effective than for instruction finetuning. We also included UltraChat [26] as generic instructions during the annealing phase.

### 3.1.3 Data Preprocessing

Our data processing pipeline closely follows [18]. In particular, we do:
1. **Text extraction**: a significant fraction of the collected data is in PDF format. We used Poppler to extract the text.
2. **Data cleaning**: extraction from PDF files creates some artifacts like page and line numbers in the middle of sentences, as well as broken lines of text, non-normalized Unicode characters, etc.

- *Text normalization*. We normalize all text using the NFKC method, available through the `unicodedata` Python package.
- *Rule-based filters*. We created regex rules for filtering commonly undesirable but commonly recurring patterns, like page and line numbers in the middle of the text, HTML tags, etc. Following [18], we found that some of the most common 10-grams in our dataset were repeated characters and whitespace and removed them.
- *Perplexity filtering*. Similarly to [18] we used a KenLM [34] model trained on a small subset of carefully cleaned legal data to filter documents with high perplexity. Concretely, we filtered any document whose normalized perplexity was larger than 1500.

3. **Text deduplication**: we used [50] to remove duplicates and near-duplicate examples from the training set. We used default parameters except for the similarity threshold, which we set to 0.5.

Finally, we packed the individual documents together to build 8192 tokens-long training examples. Documents longer than this value were chunked into several examples.

## 3.2 Instruction Data

Instruction fine-tuning is essential for making an LLM follow instructions and optimize the performance of pretrained models across a variety of tasks [81, 84, 17, 29, 26, 82]. To this end, we employ a strategic mix of general and domain-specific (legal) instructions, aimed at enhancing the model's ability to precisely interpret and execute commands, with a particular focus on legal scenarios.
**General Instructions** Our methodology for sourcing general instructions involves the integration of a diverse array of datasets, each selected to augment different aspects of the model's capabilities across various domains [14, 92]:
1. *General Instruction from UltraInteract [88]:* UltraInteract is an extensive, high-quality dataset designed to foster complex reasoning, featuring structured instructions that include preference trees, reasoning chains, and multi-turn interaction trajectories.
2. *General Instruction from Dolphin* [3]*:* This dataset provides additional conversational data, further broadening the model's exposure to diverse communication styles and contexts.

Each dataset is subjected to rigorous filtering, deduplication, and curation processes, culminating in a refined compilation of approximately 1,000,000 instructions meticulously prepared for the instruction fine-tuning phase.
**Legal Instruction Construction** For legal instructions, we synthesize dialogues and question/answer pairs that capture key legal concepts and document types to emulate legal analysis. In accordance with the model scale, we used `Mistral-54B-Instruct` for `SaulLM-54B` and `Mistral-141B-Instruct` for `SaulLM-141B`. The generation follows the recipe from [18] and begins with a three-turn sequence: (1) a user inquires about a legal document, (2) the assistant reformulates this inquiry by integrating metadata such as document type or issue date, and (3) the user asks for further explanation of the assistant's reasoning. The dialogue progressively deepens, with the assistant methodically unpacking the legal reasoning in response to increasingly nuanced questions from the user.

## 3.3 Preference Data

We enhance our models' adaptability and precision by incorporating preference data from both general and legal-specific sources [78, 62, 52, 80]. *General datasets* are UltraFeedback [20] and Orca.

---

[3] https://huggingface.co/datasets/cognitivecomputations/dolphin

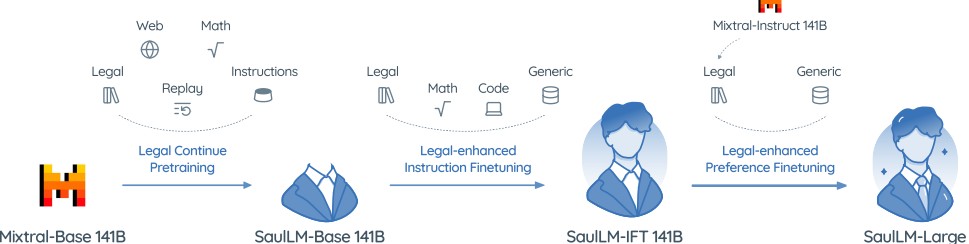

Figure 1: Domain adaptation method model for turning a Mixtral to a `SaulLM-141B`. Training involves different stages: legal domain pretraining, instruction filtering, and preference filtering.

For the *legal domain*, we employ synthetic scenarios crafted to simulate complex legal reasoning and generate accepted/rejected responses. The `Mixtral-142B-Instruct` model evaluates these responses based on factual accuracy, relevance, and logical coherence, selecting the most appropriate responses as preferred outcomes (similar to [91]).

## 4    Implementation Details & Evaluation Protocol

### 4.1    Model Selection

We used Mixtral models [40], which are built on a Transformer architecture [79] enhanced with a Mixture of Experts to improve computational efficiency and adaptability for handling extensive contexts. The `Mixtral-54B` and `Mixtral-141B` architecture respectively consists of 32 (resp. 56) layers, a model dimension of $4096$ (resp. $6144$), and a hidden dimension of $14,336$ (resp. $16384$). Although it supports a context length of up to $32,768$ (resp. $65536$) tokens, we continue pretraining on $8,192$ tokens. Extending the context length is beyond the scope of this paper. The MoE layers in Mixtral rely on 8 experts with 2 active experts selectively based on the input, efficiently utilizing computational resources and providing significant model capacity. Interestingly, Mixtral is the only model available in dual configurations (`Mixtral-54B` and `Mixtral-141B`), allowing us to evaluate the scalability of our domain adaptation approaches.

At the time of the training, Mixtral was the most powerful decoder in its class, surpassing all competitors including Llama [76, 77, 89], Yi, Qwen [7], and CroissantLLM [28] in terms of both cost-effectiveness and performance.

### 4.2    Engineering Details

**Codebase Configuration** Our training framework uses PyTorch. The integration of DeepSpeed [64] (level 3) and Flash attention [23] mechanisms enhances our training efficiency and scalability. We make our models available through the Huggingface hub [85].
**Compute Infrastructure** The computational backbone for the continuous pretraining phase of our project consists of $384$ AMD MI250 GPUs. We can reach $40\%$ GPU utilization with our implementation. For instruction fine-tuning and preference optimization, we rely on 64 AMD MI250 GPUs. Evaluation protocols are executed on a single node of AMD MI250 GPU.
**Synthetic Data Generation** For synthetic data generation, we used vLLM on a node of NVIDIA-A100, primarily due to limited support of libraries on MI250[4].

### 4.3    Training Details

The model training process is divided into three distinct phases: continued pretraining, instruction finetuning (IFT), and preference alignment using domain-specific optimization (DPO). A full schema of the pipeline can be found in Figure 1.
**Continued Pretraining** For continued pretraining, we use the AdamW [41, 47, 8] optimizer with

---

[4]VLLM is not supported on AMD-MI250 and HuggingFace's text-generation-inference had a few bugs that prevented its use.

hyperparameters $\beta_1 = 0.99$, $\beta_2 = 0.90$, and a learning rate of $2 \times 10^{-5}$. We utilize a cross-entropy loss function to optimize model predictions. The training protocol sets gradient accumulation to 4, with tailored batch sizes of 8 for `SaulLM-54B` and 4 for `SaulLM-141B`, optimizing both GPU utilization and training efficiency.

**Instruction Fine-Tuning (IFT)** IFT uses the AdamW optimizer (learning rate of $1 \times 10^{-5}$), reinitialized to reset training states and maintain training stability. We limit this phase to a single training epoch, as our experiments suggest this maximizes performance gains.

**Preference Training Using DPO** We adjust the learning rate of the AdamW optimizer to $1 \times 10^{-6}$ during DPO. Our choice of DPO over IPO [13], KTO [27] or ORPO [37] was based on preliminary experiments.

### 4.4 Evaluation Protocol

**LegalBench-Instruct** We rely on `LegalBench-Instruct` [18], which refined the prompts from `LegalBench` [31] by eliminating distracting elements and specifying a response format to enhance precision. Like LegalBench, it evaluates LLMs across six types of legal reasoning: issue-spotting, rule-recall, rule-application, rule-conclusion, interpretation, and rhetorical understanding. Grounded in American legal frameworks but applicable globally, these categories provide a comprehensive evaluation of the models' legal reasoning capabilities. This structured approach helps in accurately assessing and guiding the enhancement of LLMs within and beyond American legal contexts. We follow previous work and rely on balanced accuracy as the primary metric across all tasks.

**Massive Multitask Language Understanding (MMLU)** Previous works utilize MMLU [36], a widely-recognized benchmark, focusing on its legal-specific tasks in *international law*, *professional law*, and *jurisprudence*, with 120, 1500, and 110 examples respectively. These tasks are crucial for assessing our models' understanding and application of complex legal concepts, highlighting their proficiency in nuanced legal environments.

**Choice of Baseline & Model Naming** For our evaluation, we aim for a direct, apples-to-apples comparison of models. It is important to note that not all competing models are open source, and detailed information on their alignment procedures and instruction fine-tuning processes is not available. This lack of transparency complicates the establishment of fully equitable baseline comparisons. In what follows, we use OpenAI's `GPT-4` as of 10 May 2024, Meta's `Llama3` (the Instruct variant) and the Instruct variants of `Mixtral-54B`, and `Mixtral-141B`. Additionally, `SaulLM-54B-IFT` is the IFT version built on `SaulLM-54B-base` and `SaulLM-medium` for the DPO version based on `SaulLM-54B-IFT`. `SaulLM-large` is the final version DPO and IFT based on `SaulLM-141B`.

## 5 Experimental Results

### 5.1 Global Results

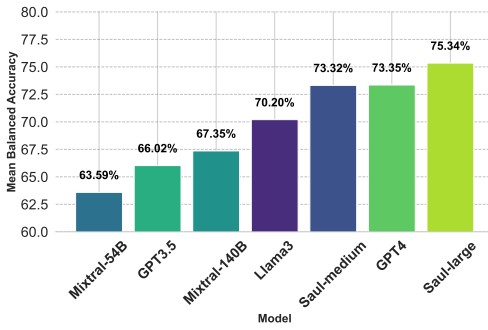

Figure 2: **Overall Results.** Comparison of SaulLM-large and SaulLM-medium with existing models.

Figure 2 presents the results of `SaulLM-large` and `SaulLM-medium` on `LegalBench-Instruct`, from which we make several observations:

**Our domain adaptation strategy is achieving strong results.** `SaulLM-medium` outperforms `Mixtral-54B`, and similar findings are observed with `SaulLM-large` compared to `Mixtral-141B`. Interestingly, domain adaptation at both the instruction tuning stage and preference alignment enables our smaller models to outperform larger ones, such as `GPT-4` and `LLama3-70B`. These results validate our approach and demonstrate that specializing the entire pipeline (*i.e.*, from continued pretraining to preference alignment) is a promising direction for improving performance in legal-related tasks.

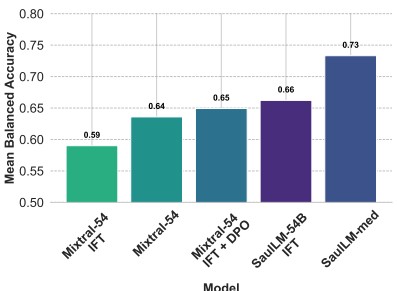

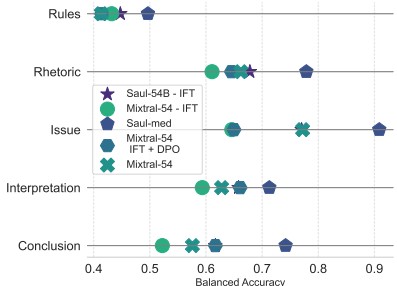

Figure 3: **Global Analysis.** Role of continued pretraining.

Figure 4: **Category Analysis:** Role of continue pretraining.

**Domain adaptation works across scales and MoE models.** The results from `SaulLM-medium` and `SaulLM-large` confirm previous findings from [18] and confirm that domain adaptation is effective across different scales, including on MoE models. Interestingly, most of the data collected for this work comes from public sources, which were likely seen during the pretraining of the base models.

**A Path Towards Stronger Models.** The results of LLama3-70B and the scalability of our methods suggest that applying the same approach to the LLama3-70B base model could lead to even better performance than our best model, `SaulLM-141B`. It is worth noting that `SaulLM-141B` has only 44B active parameters making it appealing for efficient serving.

Table 2: **Quantifying the role of DPO.** We report the percentage of tasks where the difference in performance ($\Delta$) between `SaulLM-54B` and `SaulLM-54B-IFT` is positive (resp. negative).

| Category | $\Delta \geq 0$ | $\Delta \leq 0$ |
|---|---|---|
| Conclusion | 37.5% | 62.5% |
| Interpretation | 65.1% | 34.9% |
| Rhetoric | 44.4% | 55.6% |
| Rules | 60.0% | 40.0% |
| Issue | 100.0% | - |

Table 3: **Quantifying the role of scaling.** We report the percentage of tasks where the difference in performance ($\Delta$) between `SaulLM-medium` and `SaulLM-large` is positive (resp. negative).

| Category | $\Delta \geq 0$ | $\Delta \leq 0$ |
|---|---|---|
| Conclusion | 18.2% | 81.8% |
| Interpretation | 23.7% | 76.3% |
| Rules | 25.0% | 75.0% |
| Issue | - | 100.0% |
| Rhetoric | - | 100.0% |

## 5.2 How much does continued pretraining help for the legal domain?

Previous works on domain adaptation via continued pretraining primarily focused on instruction finetuning [16, 18, 55]. In Figure 3 and Figure 4, we report the performance of `Mixtral-54B` trained with the IFT mix described in subsection 3.2 (`Mixtral-54-IFT`) and subsequently aligned using the DPO dataset (`Mixtral-54-IFT+DPO`), as described in subsection 3.3. We also compare these results to the instruct version of Mixtral (`Mixtral-54B`), as outlined in [40].

**Continuing pretraining significantly enhances model performance in the legal domain, benefiting both the IFT and DPO stages.** From Figure 3, we observe that both IFT and DPO benefit from a notable improvement (approximately $+7\%$). Interestingly, this improvement is consistent across all five categories, as shown in Figure 4.

**Adding legal data to the IFT and DPO datasets improves the model's legal capabilities.** By comparing the performance of `Mixtral-54-IFT+DPO` and `Mixtral-54`, we observe that the mix used for IFT and DPO enhanced with legal data leads to stronger legal performance than that of `Mixtral-54`, which does not publicly describe the alignment methods used. This result aligns with findings reported in [55, 18].

## 5.3 How Much Does Legal Preference Alignment Help?

Our findings from Figure 3 indicate that alignment significantly improves the results. In particular, **DPO improvement is mostly consistent across tasks and categories.** As shown in Table 2, the alignment version `SaulLM-medium` demonstrates significant improvements over the IFT version across most tasks, including conclusion, rhetoric, rules, and issue tasks. We observe, however, a drop in performance in some interpretation tasks. Upon closer examination, we found that this decline is often due to the model becoming more verbose, which causes the evaluation process to fail in correctly parsing the answers , i.e., this issue primarily arises from a benchmark limitation. Addressing model verbosity and the challenge of more reliable benchmarks is beyond the scope of this work, but it is a well-known problem identified in many concurrent studies [29]. Enhancing the evaluation process is one of the key improvements we plan to contribute to in the future.

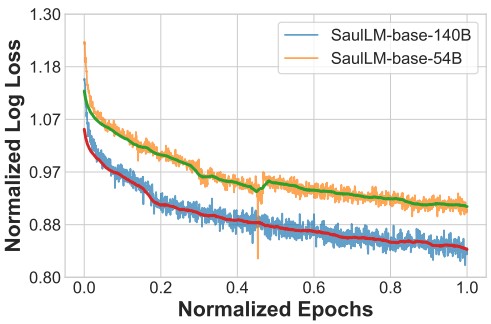

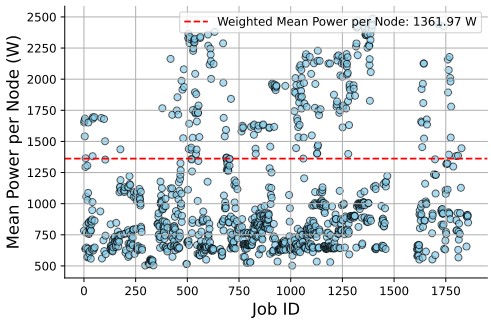

Figure 5: **Continue Pretraining Analysis.** Training loss for `SaulLM-141B-base` and `SaulLM-54B-base` over normalized epochs.

Figure 6: **Energy Consumption Analysis.** Mean Power per Node for training jobs on the ADASTRA Supercomputer.

## 5.4 Can We Achieve Further Improvements by Continuing Pretraining?

**Training longer can potentially improve the results.** Figure 5 illustrates the normalized log loss over normalized epochs for both model sizes, `SaulLM-54B-base` and `SaulLM-141B-base`. The figure presents both the raw and smoothed loss curves, which exhibit a clear downward trend throughout the training period, with no indication of saturation.

This observation suggests that continuing the pretraining process beyond the current `SaulLM-base` can lead to further improvements. The consistent decrease in loss implies that the models have not yet reached their full potential and that additional pretraining could enhance their performance further, which is consistent with findings from other works [60, 2, 51].

## 5.5 How Much Does Scaling Help?

Table 3 quantifies the impact of scaling the model and compares the performance between `SaulLM-medium` and `SaulLM-large`.

The main takeaway is that **scaling generally improves overall results, but we also observe inverse scaling on some legal tasks** [49, 43]. Unsurprisingly, for the majority of tasks across all categories, increasing the model size leads to improvements, but for tasks involving conclusion, interpretation, and rules, we observe a proportion of tasks (20%) that follow inverse scaling laws.

## 5.6 Energy Consumption

The training was conducted on Adastra, ranked 3rd in the Green500 since November 2022[5], as one of the world's most efficient machines in terms of performance per watt.

Experiments for training SaulLM were performed between February 20th and May 15th. Energy consumption for each job was meticulously tracked, and we calculated and displayed the average

---
[5] https://top500.org/lists/green500/2023/11/

power used per node for each job involved in this training in Figure 6[6]. The mean power usage ranged from 600W to 2500W, reflecting the varying utilization of the GPUs. Each node contains four MI250X GPUs, which have a theoretical Thermal Design Power (TDP) of 560W. This configuration explains the maximum consumption of 2500W during high-intensity GPU usage.

Overall, the project involves over $160,000$ hours of MI250 for debugging, continued pretraining, instruction finetuning and preference alignment. The total energy consumed was $65,480.4$kWh. Consumption is significantly lower than the typical energy requirements for full LLM training, showing that continued pretraining is an effective strategy for specializing in new LLMs while optimizing energy efficiency.

# 6    Conclusion & Limitations

## 6.1    Conclusion

This study released two new legal LLMs under the MIT license: `SaulLM-54B` and `SaulLM-141B`. They leverage the Mixtral architecture and continued pretraining on a large legal corpus. Our findings show significant advancements in processing and understanding complex legal documents. Through continued pretraining, instruction fine-tuning, and preference alignment using domain-specific optimization, we have demonstrated substantial improvements compared to `GPT-4`, `Llama3` and original Mixtral models as measured on `LegalBench-Instruct`.

## 6.2    Limitations

Our experiments suggest that the instruction finetuning and alignment processes utilized by `Mixtral-Instruct` and `Llama3` are advanced and challenging to replicate. These processes often rely on proprietary datasets and significant computational resources that are not readily available in open-source frameworks. Although both `SaulLM-54B` and `SaulLM-141B` achieve stronger performances than `Llama3` and `Mixtral Instruct` on legal benchmarks, we found that they are slightly weaker at following generic instructions.

Looking forward, we aim to continue our work on enhancing the `SaulLM` family, particularly focusing on integrating `Llama3` and improving the alignment procedure. Our goal is to improve the alignment of these models with legal tasks, refining their ability to process and understand legal language with even greater accuracy and relevance. This future work will strive to address the current limitations by developing more robust methods for instruction finetuning and alignment that are accessible to the broader research community.

## Acknowledgments and Disclosure of Funding

This research was supported by computing grants from Adastra and Jeanzay. We extend our special thanks to Michael Robert, head of CINES, for his invaluable support and confidence in our work.

Our models have been trained on ADASTRA, with minor experimentation conducted on Jeanzay. The utilization of HPC resources was made possible through the Jeanzay grants 101838, 103256, and 103298, as well as the Adastra grants C1615122, CAD14770, and CAD15031.

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

## A  Energy Analysis

Figure 7, Figure 9, Figure 8, and Figure 10 show different analysis of energy consumption while developing and training `SaulLM-54B` and `SaulLM-141B`.

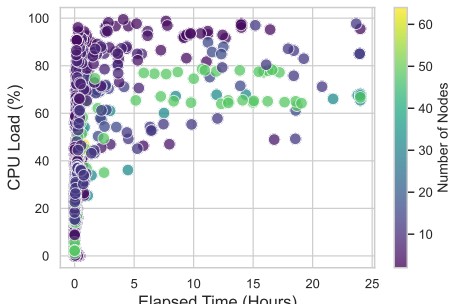

Figure 7: **Energy Analysis.** GPU Load vs Elapsed Time for Different Numbers of Nodes.

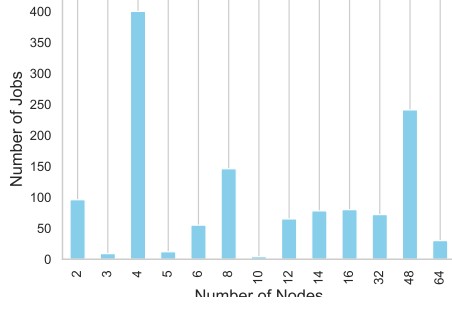

Figure 8: **Energy Analysis.** Number of Jobs vs Number of Nodes.

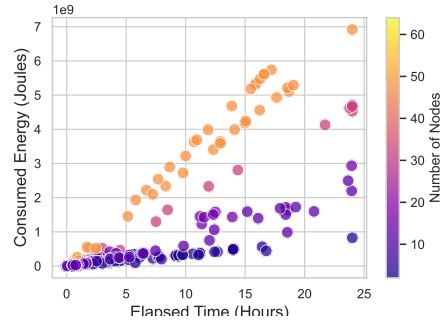

Figure 9: **Energy Analysis.** Consumed Energy vs Elapsed Time for Different Numbers of Nodes.

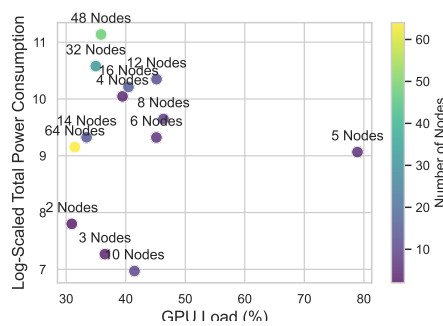

Figure 10: **Energy Analysis.** Log-Scaled Total Power Consumption vs GPU Load for Different Numbers of Nodes.

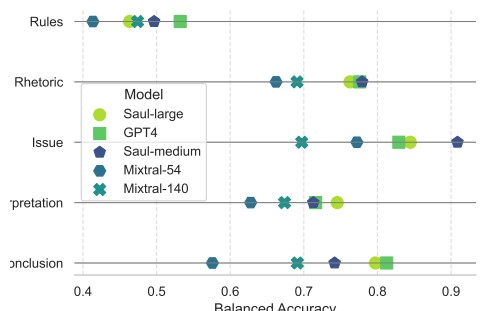

Figure 11: **Category Analysis** on `LegalBench-Instruct`.

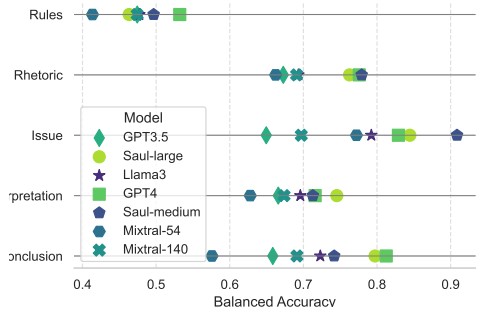

Figure 12: **Category Analysis** on `LegalBench-Instruct`.

# B Further Results on LegalBench

## B.1 Per Category Analysis

## B.2 Per Task Analysis

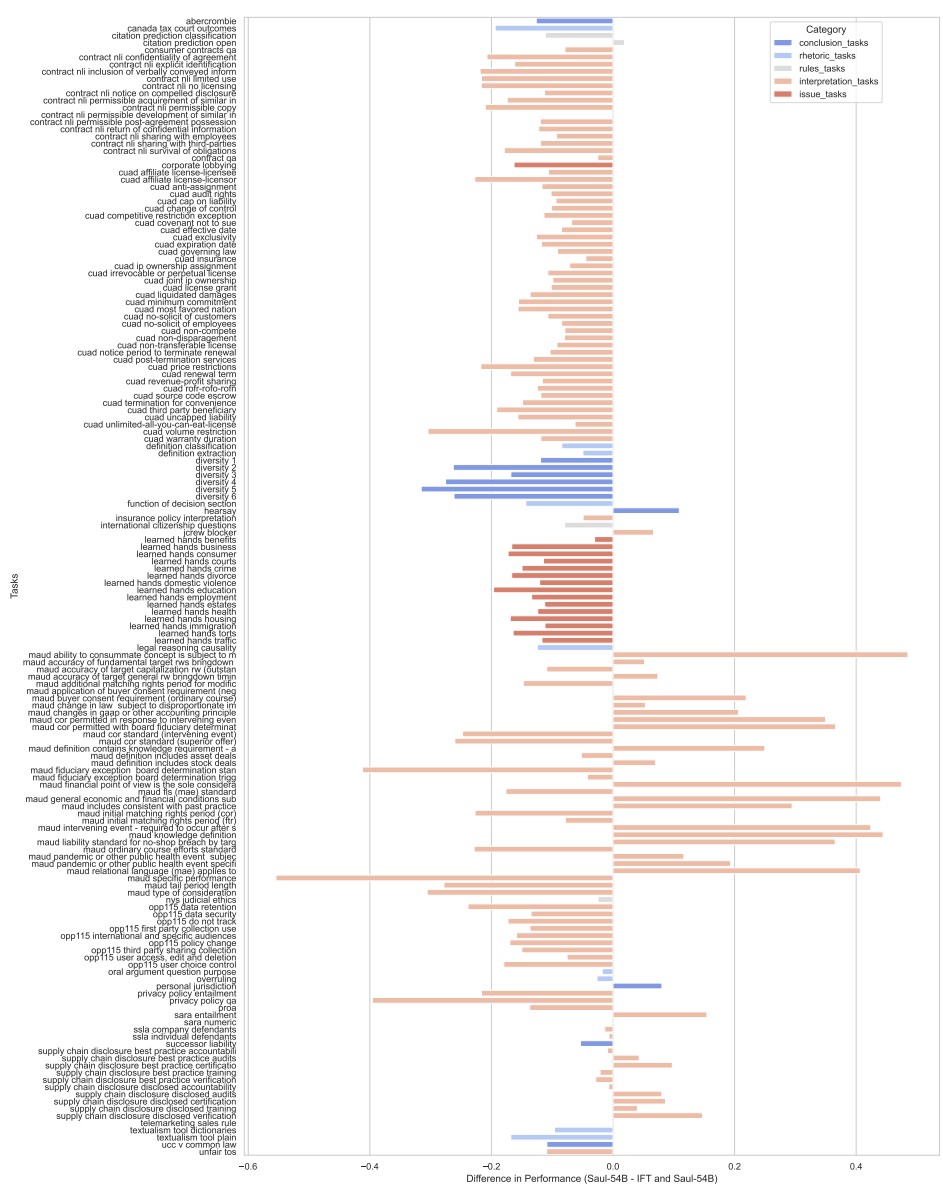

Figure 13: **Per Task Analysis:** Quantifying the role of DPO w.r.t. instruction fine-tuning only.

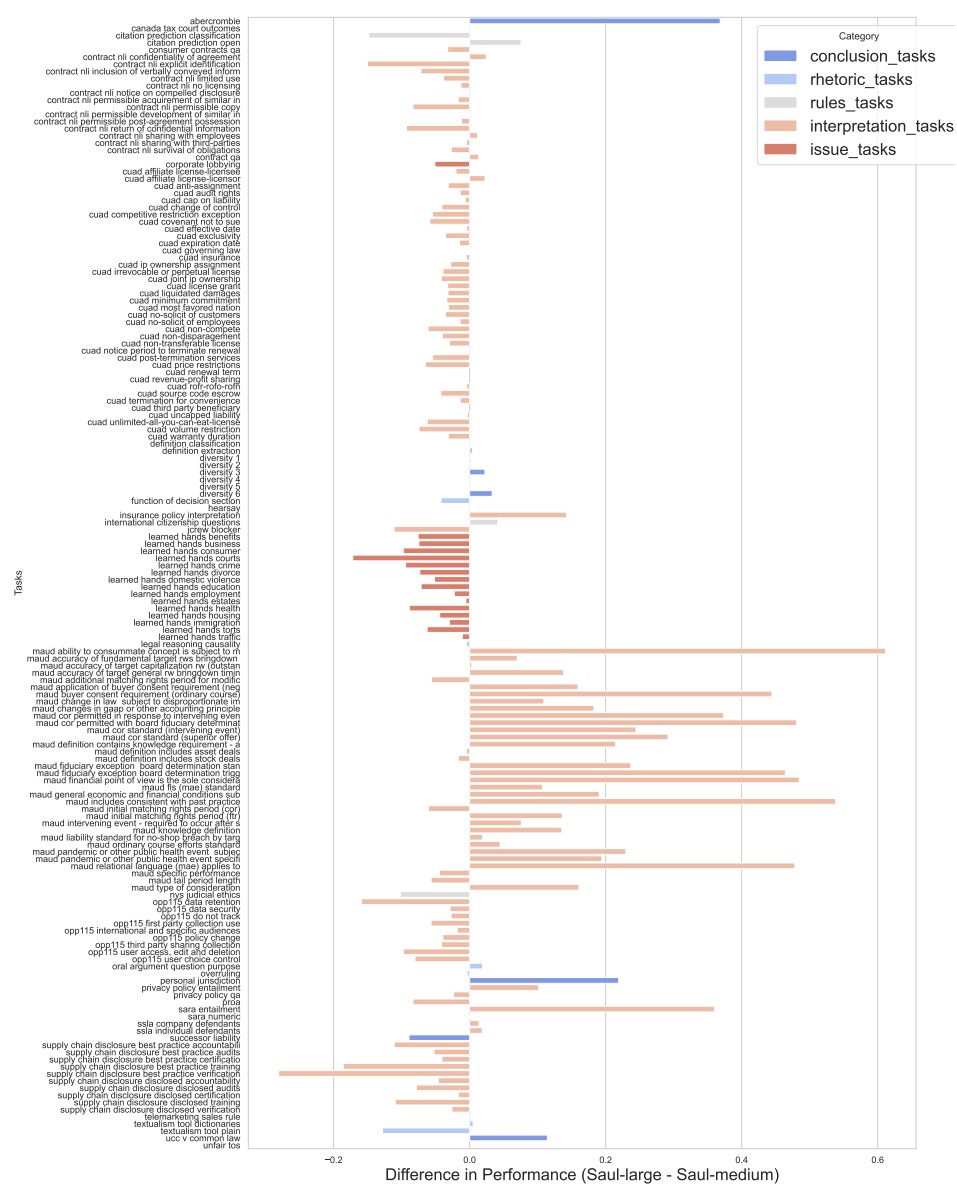

Figure 14: **Per Task Analysis:** Quantifying the role of DPO w.r.t. instruction fine-tuning only.

