# OpenReview forum: "SaulLM-54B & SaulLM-141B: Scaling Up Domain Adaptation for the Legal Domain"
_NeurIPS.cc/2024/Conference — NeurIPS 2024 poster_

### Official Review · Reviewer_jvqv · 2024-06-15

**Soundness:** 4
**Presentation:** 3
**Contribution:** 3
**Rating:** 8
**Confidence:** 3

**Summary:**

The paper reports on two new legal-specific LLMs based on Mixtral. The primary contribution is to train larger law LLMs than previously reported, using (1) an extensive dataset of legal materials, and (2) a variety of best practices in pre-training. The results indicate incremental but significant improvements on legal benchmarks.

**Strengths:**

The paper reports notable improvments in the state of the art in using LLMs for legal tasks. Most of these advantages are due to greater model size, more extensive training data for legal specialization, and engineering decisions. The results show that these techniques continue to bear fruit and have not yet reached a wall of fundamental architectural limitations. I found section 5 particularly illuminating, because it successful isolates some of the individual technical improvements to analyze which of them contribute to the performance improvement.

**Weaknesses:**

The paper's strength is also its weakness. It generates very little generalizable knowledge for understanding legal tasks, only a series of engineering improvements. There are some reasons to think that transformer-based LLMs will not be able to carry out successful human-level work without fundamental improvements in the underlying model architecture. But the research still gets us closer to the frontiers of what current LLM achitectures can do, and it is possible that similar continued improvements are all that are necessary.

**Questions:**

(See above.)

**Limitations:**

Yes, the paper is well scoped to describe what it does and doesn't do.

---

> ### Author Response · Authors · 2024-08-06
>
> **Thank you for your detailed and thoughtful feedback on our paper. We are pleased that you found our contributions significant and our paper technically strong.**
>
> We understand your concern regarding the limited generalizable knowledge for understanding legal tasks. **This limitation is partly due to the constraints imposed by the available benchmarks.** The current benchmarks are limited because collecting detailed legal benchmarks requires significant investment, **as lawyer hours are quite expensive. We will address this in our paper by adding a section on the limitations of our benchmarks.**
>
> Thank you once again for your valuable feedback.

---

### Official Review · Reviewer_pgvT · 2024-07-10

**Soundness:** 2
**Presentation:** 2
**Contribution:** 2
**Rating:** 5
**Confidence:** 4

**Summary:**

The paper introduces two large language models specialized in the legal domain with instruction-following capabilities. These models are an extension of previous work, particularly Colombo et al.'s “SaulLM-7B: A Pioneering Large Language Model for Law,” by scaling up the corpus size and the number of trainable model parameters. The model parameters are updated through continued pretraining, instruction-tuning, and alignment (RLHF, DPO, or similar methods). Additionally, the authors gather and preprocess the largest legal dataset for pretraining from various sources. They examine the impact of both model size and dataset size on the effectiveness of adapting large language models (LLMs) to the legal domain.

**Strengths:**

- The paper introduces the largest domain-specific legal language model to date, with parameters ranging from approximately 12B to 54B-141B. It curated the largest legal corpus for pretraining.

- Their results demonstrate the superiority of their models compared to state-of-the-art general-purpose language models (both open-source and closed), such as Llama 3 and GPT-4. They perform a detailed analysis of their models' performance.

- The process of training the model with all the parameters is discussed in detail. Additionally, the preprocessing of the curated corpus is clearly explained. They also depict their improvements through graphs and tables and study the impact of various factors on their results under separate headers.

- They introduce the state-of-the-art legal-domain language model, which can provide critical support to lawyers and judicial systems. They also release their model, enabling future research in the field.

**Weaknesses:**

- This paper misses ablations on the three training methods (with and without continued pretraining), ± instruct tuning, and ± alignment training. I would have liked to see where the greatest improvement in results is obtained. For example, if we do not include continued pretraining at all but only perform instruct tuning and alignment.

- The study does not include detailed results about the compared methods (Fig. 17 only compares Saul med with Saul large). Can these results be shown for the top-performing models for each training method and models of different architecture (e.g., GPT-4 and Saul large)? In the results section, they do not compare their proposed model with existing domain-specific language models like Saul-7B and legal-Flan-T5.

- The paper does not explain the process of generating synthetic data clearly. A detailed explanation of what prompts have been used to generate the data, what model has been utilized, and which generation parameters have been used is required.

- The paper discusses adding a math portion to the pretraining dataset to enhance reasoning capabilities, but the source of the data is unclear, and they don’t provide any supporting experiments to back up their claim.

**Questions:**

- What is “Score” on the y-axis of Figs. 3, 4, and 5?

- Can the authors justify the usage of only “balanced accuracy” as the primary metric used in the results associated with performance (non-energy analysis related)?

- Can the authors explain the process of preparing synthetic data for both instruction and preference fine-tuning in detail? Considering that generating synthetic data can lead to low-quality samples, did you quantitatively measure the quality of the generated data? Also, do you define any acceptance criteria for the generated data to avoid adding low-quality samples to the training corpus?

- Considering that you included data from various jurisdictions, do you evaluate the performance of the model across different jurisdictions?

- Based on the results presented in Figure 6, do you have any explanation or assumption as to why Mixtral-IFT is performing worse than Mixtral?

- In the conclusion, you claim that “we have demonstrated substantial improvements compared to GPT-4.” The mean balanced average of GPT-4 and the proposed model are pretty close. Did you perform any significance tests to show that the difference is statistically significant?

- The authors clearly indicated that the experiments are limited due to the proprietary nature of the datasets used to train Llama and Mixtral. Have the authors looked into open-source and not just open-weight models? For example, Olmo: https://arxiv.org/abs/2402.00838.

**Limitations:**

Yes.

---

> ### Author Response · Authors · 2024-08-06
>
> We thank the reviewer for their review. We are glad they found our paper is well written and our contribution extremely valuable.
>
> Below is the response to the concerns/questions:
>
> **About the additional ablations**, they are already reported in the paper. See general comments.
>
> **Adding comparison to GPT-4.**
>
> We acknowledge the request for more detailed comparisons similar to those depicted in Figures 17 and 18, which illustrate the relative improvement from DPO with respect to instruction fine-tuning and the effects of model scaling within the relying on the Mixtral backbone.
> In the revised version, we extend these with a comparison to GPT-4, although emphasizing that GPT-4's size—1.5 trillion parameters—makes direct comparisons challenging. Our focus remains on domain adaptation of base models, ensuring apple-to-apple comparisons, rather than contrasting models of substantially different sizes.
> We have included results in Table 3 and Figure 5 and note that our models compare favorably with GPT4.
>
> **On the addition of SaulLM-7B and Legal Flan T5:** See general comment
>
> **Synthetic data generation:** see General Comment.
>
> **Addition of math data:** We performed an ablation study on a 7B model (not reported in the paper). Below are the results from the reviewers:
>
> | Models               | Size | Results |
> |----------------------|------|---------|
> | Mistral 7B + pretraining (without math) + IFT   | 7B   | 0.612    |
> |  Mistral 7B + pretraining (without math) + IFT    | 7B   | 0.628    |
>
> In the revised version we did report the results in the appendix.
>
> **About the reviewer’s questions:**
>
> **Score refers to balanced accuracy.** We will improve the label.
>
> **Balanced accuracy** was originally used by the Legalbench authors (see section 5.1.3 of their paper). We reused their code for comparizon.
>
> **See the general comment for the synthetic data.**
>
> **We will add examples of generation to the appendix.**
>
> **Comparison on other jurisdictions.** For this study, we focused on LegalBench, which is emerging as a standard benchmark. While we plan to expand this benchmark to include other jurisdictions, such an extension requires significant time and financial resources; therefore, we did not evaluate other jurisdictions in this iteration. However, we conducted manual tests using concepts from European law, such as "imprevision", "basis of the bargain" or "abuse of dominant position", and found that the responses were significantly improved compared to the original model from Mixtral. We intend to explore this further and provide a detailed evaluation in a future paper. **We plan to add these examples in the revised version of the paper.**
>
> **IFT v.s. Mixtral:** We believe that the Mixtral models align more closely with DPO, whereas the IFT model does not. This alignment discrepancy may represent a key difference between the models. Due to the limited information available about their training processes, we conducted an ablation study in Section 5.2 to further dissect and understand these different factors. The reviewer may notice that once aligned with legal synthetic data the IFT + DPO outperforms Mixtral.
>
> **About the significance of the comparison with GPT4.** LegalBench reports over 90k samples by relying on the methodology of (D Card et al. EMNLP2020), we see a power of the test of over 95%. We have to restate that the goal of the paper is to study domain adaptation and thus the main comparison should be w.r.t. the mixtral models.
>
> **About Olmo.** We are excited about the Olmo initiative and plan to reference it in our upcoming paper. Given our interest in working with models larger than 7 billion parameters, the Mixtral family was our preferred choice.
>
> **We hope that we have addressed the reviewers’ concerns and hope they will increase their grades.**

---

### Official Review · Reviewer_LHKy · 2024-07-13

**Soundness:** 3
**Presentation:** 3
**Contribution:** 3
**Rating:** 7
**Confidence:** 4

**Summary:**

- The paper introduce two LLMs (at different sizes) specialized for law. These models have been adapted for law through continued pretraining, specialized legal instruction following, and a “legal alignment” process
- The paper studies the tradeoffs of domain adaptation at this scale and presents results for these models.

**Strengths:**

- To the best of my knowledge, this is the first study of domain adaptation for legal LLMs at this scale (model size and training corpora size). This makes it an extremely valuable contribution.
- The paper is extremely clear and well-written.
- The experiments seem thorough and well motivated.

**Weaknesses:**

- The paper is sparse on some key details related to the datasets used to train the model.
    - For the preference data, what are the synthetic scenarios designed? How were they constructed? What were the Mixtral prompts used to evaluate them?
    - For the legal instruction data, what were the legal documents initially chosen? What did the conversations look like?
- Examples of both instruction and preference data in the Appendix would be very helpful to readers.
- The ablations are extremely helpful but also difficult to read, in part because they’re mixed in with baselines. It would be nice if the authors could present a table showing the progression of performance between: Mixtral-54B, + continued pretraining on the legal corpora, + IFT, + legal alignment. I think some–but not all–of these in Figure 6?

**Questions:**

- The LegalBench website asks users of the benchmark to cite a collection of papers, not merely the original LegalBench paper. Since LegalBench-Instruct looks like a small modification to LegalBench, do you plan to cite that original collection of papers as well? Also the LegalBench citation seems wrong, and not to the actual LegalBench paper (https://arxiv.org/abs/2308.11462)
- Do the authors plan on releasing these datasets?
- Do the authors plan on releasing intermediate checkpoints for model training?]
- Once concern in relying on LegalBench-Instruct is that the prompt format chosen favors Mixtral models over other types of models (e.g., Llama-3, GPT, etc.). Do the authors have a sense of how sensitive the Saul models are to the prompt format?
- It would be very nice to see some examples of generations in the Appendix!

I’ll increase my score if the additional details are added to the paper.

---

> ### Author Response · Authors · 2024-08-06
>
> **We thank the reviewer for their review we are glad they acknowledged that our paper is well written and find that our contribution is extremely valuable.**
>
> Below is the response to the concerns/questions:
>
> **About the generation of the post-training datasets and examples.** See the general comment.
>
> **Ablation:** See general comment on section 5.2
>
> **Missing citations:** have been identified as an oversight on our part and have now been corrected in the revised version. We will include references to LegalBench as well as all pertinent datasets, specifically citing Guha et al., 2023; Koreeda and Manning, 2021; Hendrycks et al., 2021; Wang et al., 2023; Wilson et al., 2016; Zheng et al., 2021; Zimmeck et al., 2019; Ravichander et al., 2019; Holzenberger and Van Durme, 2021; Lippi et al., 2019.
>
>
> **On the prompt format:** In our study, we adhered strictly to the original prompt formats from the LegalBench paper, relying solely on the prompts from the dataset itself for generating responses. It's important to clarify that our primary objective is to examine domain adaptation, which necessitates careful adjustments. While we also compare our results with GPT-4, it's important to note that this comparison may not be entirely equitable; GPT-4's model size is substantially larger—approximately 10 times that of a 140B model and 30 times that of a 54B model. Despite testing slight variations in prompt formatting, we observed minimal impact (less than 0.2%) on our models across approximately 100,000 total samples, leading us to report outcomes using the standard version.
>
> **Regarding release artifacts, upon acceptance, we plan to release both the curated training dataset and the checkpoints along with optimizer states for seamless training resumption.**
>
> **We hope that we have addressed the reviewers’ concerns and hope they will increase their grades.**

---

> > ### Comment · Reviewer_LHKy · 2024-08-12
> >
> > Thanks for the additional details! I'll raise my score.

---

### Official Review · Reviewer_iZW5 · 2024-07-20

**Soundness:** 3
**Presentation:** 3
**Contribution:** 3
**Rating:** 5
**Confidence:** 4

**Summary:**

The paper introduces SaulLM-54B and SaulLM-141B, two large language models specifically designed for the legal sector. These models utilize the Mixtral architecture and are developed through extensive domain adaptation strategies, including continued pretraining on a large legal corpus, instruction fine-tuning, and preference alignment. The models incorporate synthetic data to enhance their capabilities in legal text interpretation and achieve state-of-the-art performance on LegalBench-Instruct. The study emphasizes the scalability of domain-specific adaptation and the potential benefits of larger model sizes for legal applications.

**Strengths:**

1. SaulLM models are the first open-source legal LLMs with a larger model size, utilizing continual pretraining, instruction tuning, and RLHF.

2. The release of these models under the MIT License facilitates reuse and collaborative research in legal NLP.

**Weaknesses:**

1. The techniques used, such as continual pretraining, instruction tuning, and preference alignment, are not novel compared to existing studies like SaulLM and LLMs in other domains.

2. There are issues with the license, privacy, and quality of data. The paper should explicitly illustrate the license of each used data source. The quality of synthetic legal instruction tuning data and preference data is uncertain.

3. The evaluation is not comprehensive. There is no ablation study to illustrate the effectiveness of continual pretraining, such as comparing the performance of Mixtral-54B and SaulLM-54B-base. It would be better to include other LLMs in the legal domain as baselines.

4. In section 5.2, the explanation of "How much does continued pretraining help for the legal domain" is confusing. The continual pretraining previously referred to unsupervised continual pretraining with a next-token generation task on large-scale legal data, but in this section, it seems to mean IFT+preference alignment.

**Questions:**

On page 7, it is mentioned: "The results of LLama3-70B and the scalability of our methods suggest that applying the same approach to the LLama3-70B base model could lead to even better performance than our best model, SaulLM-141B." Which figure's results support this illustration?

**Limitations:**

Yes

---

> ### Author Response · Authors · 2024-08-06
>
> We thank the reviewer for their review we are glad they acknowledged the usefulness of our paper and our contribution to the community with these models.
>
> Below are the answers to the reviewer's comments:
> **There are no issues with the licenses:** The licenses permit commercial use for both the pretraining and instruction finetuning data. Below is a breakdown of the licenses:
>
> | Source Name                                 | Tokens (B) | License                        |
> |---------------------------------------------|------------|--------------------------------|
> | FreeLaw Subset from The Pile                | 15         | MIT license                    |
> | EDGAR Database                              | 5          | Apache 2                    |
> | English MultiLegal Pile                     | 50         | CC-By-SA-4.0                   |
> | English EuroParl                            | 6          | Public Domain                  |
> | GovInfo Statutes, Opinions & Codes          | 11         | Open government license        |
> | Law Stack Exchange                          | 0.019      | CC-By-SA-4.0                   |
> | Comm Open Australian Legal Corpus           | 0.5        | CC-By-4.0                   |
> | EU Legislation                              | 0.315      | CC-By-4.0                   |
> | UK Legislation                              | 0.190      | Open government license        |
> | Court Transcripts                           | 0.350      | CC-By-ND international 4.0       |
> | UPSTO Database                              | 4.7        | CC-By-SA-4.0                   |
> | Web Data (legal)                            | 400        | ODC-BY                   |
>
> **Upon acceptance, we plan to include all the licenses in the appendix.**
>
> **The doubt The quality of synthetic legal instruction tuning data and preference data is uncertain.** We believe that the quality of the post-training is demonstrated by the results.
>
> **Including other Legal LLMs as a baseline:** See general comment.
>
> **Clarification of section 5.2.**  See general comment.
>
> **About the Claim:** On page 7, it is mentioned: "The results of LLama3-70B and the scalability of our methods suggest that applying the same approach to the LLama3-70B base model could lead to even better performance than our best model, SaulLM-141B."
>
> The intention behind this statement is to highlight that the LLama3-Instruct results demonstrate stronger performance compared to Mixtral 7X22B (see Fig. 2). Based on this, we believe that applying the same methods could yield even stronger results. We acknowledge that this sentence may be confusing, and we will remove it in the updated version of the paper to avoid any ambiguity.
>
>
> **We hope that we have addressed the reviewers’ concerns and hope they will increase their grades.**

---

### Author Response · Authors · 2024-08-06

The reviewers positively recognized the significance of our paper (jvqv), noting its application in the legal domain (pgvt), scale (LHKy, pgvt), results (jvqv, pgvt), and openness (iZW5, pgvt). They also find the paper clear and well-written (jvqv).  Below, we answer the common questions of the reviewers about the choice of the baseline (pgvT, iZW5), about synthetic data (pgvT, LHKy) and about the ablation study (pgvT, LHKy).

**Ablation Study:  All results asked by the reviewers are already provided in Figures 6 and 7**
5.2.1 allows quantifying the effect of pretraining, IFT and DPO. All results asked by the reviewers are already provided in Figures 6 and 7. Below is a mapping of the model name referenced in the paper as well as a model description:

| Model Description                                      | Reference Name in the paper       | Results |
|--------------------------------------------------------|----------------------|---------|
| Mixtral (original) + IFT (our IFT)                     | Mixtral 54-IFT       | 0.59    |
| Mixtral Instruct (original from Mixtral)               | Mixtral 54B          | 0.64    |
| Mixtral (original) + IFT (our IFT) + DPO (our DPO)     | Mixtral 54-IFT + DPO | 0.65    |
| Mixtral (original) + Continued Pretraining + IFT (our IFT) | SaulLM-med 54 IFT    | 0.66    |
| Mixtral (original) + Continued Pretraining + IFT (our IFT) + DPO (our DPO) | SaulLM-med | 0.73

This allows the reviewer to quantify the impact of various improvements, including DPO, IFT, and continued pretraining. **A clarification has been made** in the revised version of the paper to avoid any ambiguity.

---

> ### Author Response · Authors · 2024-08-06
>
> **Clarification on additional Baseline and the main claim of the papers.**
>
> We appreciate the suggestion to include other legal LLMs as baselines. However, previous legal LLMs, such as SaulLM, are relatively small in scale (up to 7B parameters) and Legal-Flan-T5, is not available (not released by the authors) for comparison. For completeness, we will report the results of SaulLM:
>
> | Models               | Size | Results |
> |----------------------|------|---------|
> | SaulLM 7B-Instruct   | 7B   | 0.61    |
> | Mistral 7B v2        | 7B   | 0.52    |
> | Mixtral 54B          | 54B  | 0.64    |
> | SaulLM-Base          | 54B  | 0.73    |
> | Mixtral 140B         | 140B | 0.65    |
> | SaulLM-Large         | 140B | 0.75    |
>
> **This comparison may not be entirely fair due to differences in model size and the pretraining regime of our paper.**

---

> ### Author Response · Authors · 2024-08-06
>
> **About the post-training datasets.**
>
> **For the instruction finetuning dataset:** We follow the SaulLM-7B recipe that relies on Ultrachat methodology as described in their original paper. Concretely, we generate detailed discussions on essential legal skills across various document types using a Mistral-140B-instruct to convert legal texts with metadata into structured dialogues (Ding et al., 2023). The conversation starts with three steps: (1) a user request about the document, (2) the assistant rephrases metadata like document type and date, and (3) the user prompts for deeper reasoning. Further exchanges involve the user asking targeted questions to understand the assistant's logic, with the assistant providing thorough explanations. We removed the 30% conversation that has the higher number of duplicated n-grams (computed independently for the user and assistant) which we found good enough.
>
> **Example:**
>
> 1. **User:**
>    "Here is an extract from a legal contract: 'Except as otherwise set forth in this Debenture, the Company, for itself and its legal representatives, successors and assigns, expressly waives presentment, protest, demand, notice of dishonor, notice of nonpayment, notice of maturity, notice of protest, presentment for the purpose of accelerating maturity, and diligence in collection.' Tell us which label would you use to best describes this extract?”
>
> 2. **Assistant:**
>    "The most relevant label would be 'Waivers'."
>
> 3. **User:**
>    "Explain your reasoning."
>
> [....]
>
> 8. **Assistant:**
>    "These elements typically serve as enforcement mechanisms in debt repayment. Notices inform debtors of obligations or issues; demands formally request action or payment; protests document objections to non-payments; and presentments involve the physical delivery of financial documents. Waiving these steps simplifies and potentially expedites the debt handling process by removing formalities that might otherwise escalate disputes or delays."
>
>
> **For the alignment dataset,** we start from a dataset where we ask the LLM to generate a question. From this question and the input sample, we generate 2 responses from the LLM. Then, we ask a judge to rank the two answers (see description in the paper: ). The first is the chosen and the second is the rejected. We rely on Mixtral 8x22B as described in Fig. 2. This is similar to the Nemotron paper (Nvidia Team 2024).
>
> Example:
>
> **Question:** Which category best describes the following extract from a legal contract?
>
> "Except as otherwise set forth in this Debenture, the Company, for itself and its legal representatives, successors and assigns, expressly waives presentment, protest, demand, notice of dishonor, notice of nonpayment, notice of maturity, notice of protest, presentment for the purpose of accelerating maturity, and diligence in collection."
>
> LLM 1: Waivers
>
> - **Reasoning:** This label is most appropriate because the [...]
>
> LLM 2: Modifications
>
> - **Reasoning:** One might also consider this as a  'Modifications' because the waiver of traditional debt collection rights [..]
>
> LLM as a Judge: Waivers
>
> Reasoning: The extract explicitly states that the company "expressly waives" several rights, [....]
>
> **We have added more details in the paper’s appendix and we plan to release the datasets upon acceptance to clarify and support the reader. We will also include examples in the appendix.**

---

### Decision · Program_Chairs · 2024-09-25

**Decision:**

Accept (poster)

**Comment:**

The paper presents SaulLM-54B and SaulLM-141B, two large language models tailored for legal applications. Built on the Mixtral architecture, these models are refined using comprehensive domain adaptation techniques, such as continued pretraining on a vast legal text corpus, instruction fine-tuning, preference alignment and synthetic data generation. The models achieve state-of-the-art results on the LegalBench-Instruct benchmark. The research highlights the effectiveness of scaling domain-specific adaptations and the advantages of larger model sizes in enhancing legal language processing.

The main strength of the paper is in demonstrating strong domain adaptation results for a specific and important domain, for large-size LLMs. The main limitation, when it comes to a scientific conference, is the use of well-known and well-explored techniques (even though they have not been shown to provide such strong results for domain adaptaiton of LLMs of this size). Yet, the engineering result is strong and novel. The divergence in reviewer scores is mostly with respect to how they weigh the obvious pros and cons mentioned above. I am leaning towards accept.